# The evolution of process-based hydrologic models: Historical challenges and the collective quest for physical realism

*Martyn P. Clark[1], Marc F. P. Bierkens[2], Luis Samaniego[3], Ross A. Woods[4], Remko Uijlenhoet[5], Katrina E. Bennett[6], Valentijn R. N. Pauwels[7], Xitian Cai[8], Andrew W. Wood[1], and Christa D. Peters-Lidard[9]*

[1]Research Applications Laboratory, National Center for Atmospheric Research, Boulder, CO 80301, USA

[2]Faculty of Geosciences, Utrecht University, 3508 TC Utrecht, The Netherlands

[3]UFZ-Helmholtz Centre for Environmental Research, Leipzig, 04318, Germany

[4]Department of Civil Engineering, University of Bristol, Bristol, BS8 1TR, UK

[5]Hydrology and Quantitative Water Management Group, Wageningen University, 6700 AA Wageningen, The Netherlands

[6]Los Alamos National Laboratory, Los Alamos, NM 87545, USA

[7]Department of Civil Engineering, Monash University, Victoria 3800, Australia

[8]Department of Civil and Environmental Engineering, Princeton University, Princeton, NJ 08544 USA

[9]Earth Sciences Division, NASA Goddard Space Flight Center, Greenbelt, MD 20771, USA

## Abstract

*The diversity in hydrologic models has historically led to great controversy on the "correct" approach to process-based hydrologic modeling, with debates centered on the adequacy of process parameterizations, data limitations and uncertainty, and computational constraints on model analysis. In this paper, we revisit key modeling challenges on requirements to (1) define suitable model equations, (2) define adequate model parameters, and (3) cope with limitations in computing power. We outline the historical modeling challenges, provide examples of modeling advances that address these challenges, and define outstanding research needs. We illustrate how modeling advances have been made by groups using models of different type and complexity, and we argue for the need to more effectively use our diversity of modeling approaches in order to advance our collective quest for physically realistic hydrologic models.*

## 1    Introduction

The research community exhibits great diversity in its approach to hydrologic modeling, with different models positioned at different points along a continuum of complexity. Models can be defined both in terms of process complexity, i.e., to what extent do different models explicitly represent specific processes; and spatial complexity, i.e., to what extent do different models explicitly represent details of the landscape and the lateral flow of water across model elements. Such model diversity has led to great community debates on the "correct" approach to process-based hydrologic modeling [Wood et al. 1988; Grayson et al. 1992b, 1992a; Famiglietti and Wood 1995; Reggiani et al. 1998; Beven 2002; Sivapalan et al. 2003; Maxwell and Miller 2005; Wood et al. 2011; Beven and Cloke 2012; Wood et al. 2012], with the debate centered around issues of the adequacy of process parameterizations, data limitations and uncertainty, and computational constraints on model analysis.

This synthesis paper is an outcome of the Symposium in Honor of Eric F. Wood, Observations and Modeling across Scales, held June 2-3, 2016 in Princeton, New Jersey, USA. The purpose of this paper is to revisit the historical debates on process-based hydrologic modeling and ask the following question: *How can we combine different perspectives on hydrologic modeling to advance the quest for physical realism*? [Kirchner 2006; Clark et al. 2016]. Specifically, we focus attention on the three fundamental questions that were posed by Freeze and Harlan [1969] in their seminal "blueprint" for a physically-based hydrologic response model:

1. Are physically based mathematical descriptions of hydrologic processes available? Are the interrelationships between the component phenomena well enough understood? Are the developments adaptable to a simulation of the entire hydrologic cycle?

2. Is it possible to measure or estimate accurately the controlling hydrologic parameters? Are the amounts of necessary input data prohibitive?

3. Have the earlier computer limitations of storage capacity and speed of computation been overcome? Is the application of digital computers to this type of problem economically feasible?

We posit that these questions, published almost fifty years ago, are very relevant today and nicely frame the debates on process-based hydrologic modeling.

We organize the paper around the three questions posed by Freeze and Harlan, on (1) model structure; (2) model parameter values; and (3) model execution (computing). We discuss these modeling challenges separately, while recognizing that these modeling challenges are strongly interdependent (e.g., a complex model structure may have large computing demands, restricting the extent to which it is possible to explore alternative model parameter sets). We will touch on these interdependencies in the individual sections of the paper.

For each question posed by Freeze and Harlan we define the major research challenges, and we provide examples of different ways that the community has risen to meet these challenges, considering modeling approaches of varying complexity. We do not mean to provide a comprehensive review; rather, we present possible solutions to outstanding modeling problems, focusing attention on the research sphere of Eric F. Wood. Our overall intent in writing this paper is to demonstrate how diverse hydrologic modeling approaches advance the collective quest for physically realistic hydrologic models, and to define additional research that is necessary to further advance process-based hydrologic models.

## 2  Model structure

### 2.1  Modeling challenges

The first question posed by Freeze and Harlan [1969] focuses on the adequacy of the mathematical descriptions of system of interest. Such mathematical descriptions define the structure of a model. They include both the equations used to parameterize individual processes as well as the interactions among processes and across scales.

A major research challenge addressed by Eric F. Wood is the problem of scaling, or closure [Wood et al. 1988; Blöschl and Sivapalan 1995; Reggiani et al. 2001; Beven 2006], i.e., how best to represent the influence of small-scale heterogeneities on large-scale fluxes, and how best to represent interactions among processes and the connectivity of water across the landscape. The scaling challenge is ubiquitous. For example, Mahrt [1987] demonstrates

how localized areas of instability can dominate large-scale energy fluxes; Scott et al. [2008] demonstrate that transpiration from narrow riparian corridors in arid regions is much greater than the local precipitation; Seyfried et al. [2009] demonstrate that deep snow drifts produce local runoff "hotspots" that generate a disproportionate amount of the catchment runoff; Tromp-van Meerveld and McDonnell [2006a, 2006b] demonstrate that the water stored in bedrock depressions must be raised to a sufficient level in order to connect bedrock depressions and generate hillslope outflow. The community has risen to meet these scaling challenges in very different ways – different models use very different sets of equations to describe the large-scale manifestation of spatial heterogeneity, process interactions, and connectivity.

The different solutions to the scaling/closure problem can be distinguished by the extent to which the effort is focused on developing new large-scale flux parameterizations or numerically integrating the small-scale heterogeneities across space. Such differences are perhaps best illustrated by considering the different approaches used to simulate the transmission of water through catchments. In bucket-style rainfall-runoff models—at the simplest end of the complexity continuum—the large-scale transmission of water is often defined as a linear (or near-linear) function of water storage (e.g., see the synthesis in Clark et al. [2008] and the recent review by Hrachowitz and Clark [2017]). Such large-scale closure relations implicitly represent the small-scale heterogeneity of flow paths, including the localized areas of high conductivity (e.g., macropores) that dominate the large-scale response [Beven and Germann 1982; McDonnell 1990]. By contrast, the more complex 3D variably saturated flow models typically use small-scale closure relations [Maxwell and Miller 2005; Rigon et al. 2006], where unsaturated hydraulic conductivity is defined as a highly non-linear function of soil moisture [e.g., Van Genuchten 1980]. These 3D models compute large-scale fluxes by spatially integrating the small-scale heterogeneities [Maxwell and Kollet 2008; Kollet et al. 2010]. The differences in solutions to the scaling problem are not mutually exclusive, as many models include a mix of small-scale and large-scale flux parameterizations (e.g., VIC uses a large-scale parameterization of infiltration, yet relies on small-scale equations to simulate the storage and transmission of water through the upper portion of the soil matrix).

When viewed in this way, the different solutions to the scaling/closure problem can be shared among different modeling groups that employ very different modeling approaches. To explain this perspective, consider the inequality that describes ideal relationships between the model resolution and the length scale of resolved and unresolved processes [Wood et al. 1988]

$$l \ll D \ll L \tag{1}$$

where $l$ is the length scale of the rapidly varying hydrologic response, $L$ is the length scale of the slowly varying quantities, and $D$ is the length scale of the model element (note the assumption that the spatial scale of processes below the grid resolution is clearly separated from the spatial scale of processes above the model resolution; a condition that is rarely achieved in practice [Fan and Bras 1995]). Critically, equation (1) requires that processes below the length scale of the model element must be represented implicitly (e.g., through large-scale flux parameterizations) and processes above the length scale of the model element must be represented explicitly (e.g., through numerical integration over spatially distributed model elements).

The trend towards "hyper" resolution land models [Wood et al. 2011], e.g., 1km or 100m over large geographical domains, emphasizes the need for general parameterizations of

hydrological processes at this scale. However, this is still an unsolved problem: we do not have firm evidence that the structure and parameter values of our element-scale equations correspond to hydrologic reality at those scales. One of the most important causes of this difficulty is the spatial heterogeneity in the initial and boundary conditions, and in the material properties of the medium. This heterogeneity occurs at multiple spatial scales, and has multiple physical causes [Seyfried and Wilcox 1995]. The multiple scales of heterogeneity are manifest as multiple dominant processes [Grayson and Blöschl 2001], and also as processes without a well-defined spatial scale (e.g. preferential flow in the snowpack, on the land surface, in the subsurface). These problems cannot be solved solely by numerical integration across space. The next section summarizes recent advances in developing large-scale flux parameterizations, and in effectively resolving dominant processes.

### 2.2 Modeling solutions

The common challenge of developing large-scale flux parameterizations has been addressed in a number of ways. One class of methods is statistical-dynamical flux parameterizations, where large-scale fluxes are defined based on probability distributions of sub-grid or sub-element model state variables. For example, area-average infiltration can be parameterized based on probability distributions of water table depth [Beven and Kirkby 1979; Sivapalan et al. 1987] or on probability distributions of soil moisture [Moore and Clarke 1981; Wood et al. 1992]. Statistical-dynamical approaches are also used to parameterize the impact of frozen soils on area-average infiltration [Koren et al. 1999] and the impact of spatial variability of snow on area-average energy fluxes [Luce et al. 1999; Liston 2004; Clark et al. 2011a]. Another class of methods consists of scale-dependent parameterizations, where new flux parameterizations are defined directly at the scale of interest. Examples of this class of methods include the empirically derived storage-discharge relationships described earlier, where the large-scale transmission of water is often defined as a linear (or near-linear) function of water storage [Ambroise et al. 1996; Clark et al. 2008; Fenicia et al. 2011; Brauer et al. 2014]. Similarly, large-scale stability corrections, used in computations of land-atmosphere energy fluxes, implicitly represent the impact of local pockets of instability on large-scale fluxes [Mahrt 1987]. There is a strong need to synthesize, evaluate, and compare these large-scale parameterizations, in order to improve the physical realism of hydrologic models [Clark et al. 2011b; Clark et al. 2015b; Clark et al. 2016].

Statistical-dynamic flux parameterizations rely on the assumption that the model scale $D$ is large compared to the length- or time-scale of the heterogeneity of hydrological response $l$. In other words, the size of a model element is large compared to the scale-of-fluctuation [Rodríguez‑Iturbe 1986] or the integral-scale [Dagan 1994] of the underlying process. In that case, univariate probability density functions can be used that, when spatially, temporally or probabilistically integrated, result in small variance representative parameters at the scale of the model elements that do not depend on the model state (called full closure). However, it becomes more difficult to define scale-aware parameterizations if $l$ and $D$ are comparable in scale. Here, much can be learned from the upscaling research that has been done in stochastic subsurface hydrology to derive representative hydraulic conductivities at the scale of model blocks [see Sánchez-Vila et al. 1996 for a review]. These approaches can be distinguished into two main categories [Bierkens and Van der Gaast 1998]: direct upscaling, whereby the spatial statistics, i.e. mean and spatial covariance, of the block-scale hydraulic conductivity are directly derived from integrating the small scale spatial statistics, and indirect upscaling where the hydraulic conductivity is first stochastically simulated or interpolated at the smallest scale and then upscaled by non-

linear averaging. Direct methods work best for heterogeneity that can be described by multi-Gaussian random functions. However, numerical integration across space may be necessary if the heterogeneity is more organized or of larger complexity. It is important to notice however, that full closure is often not possible, resulting in representative parameterizations that change with the model state.

The challenge of effectively resolving dominant processes has also been tackled in different ways. While one tactic is to simply discretize the domain into the highest resolution grid that modern computers allow (the numerical integration across space described above) [Freeze and Harlan 1969; Maxwell et al. 2015], this approach constrains capabilities to extensively experiment with alternative model configurations and to characterize model uncertainty [Beven and Cloke 2012; Wood et al. 2012]. Hence, for practical reasons, the challenge of spatial integration is commonly met using concepts of hydrologic similarity, often implemented at multiple levels of granularity within the same model. At a fine level of granularity, Wang and Leuning [1998] make separate stomatal conductance calculations on sunlit and shaded leaves to improve scaling from the leaf to the canopy. Similarly, Swenson and Lawrence [2012] make separate energy balance calculations over snow covered and snow free terrain to improve estimates of large-scale energy fluxes. At the system scale, many models spatially integrate across discrete landscape types to capture the large-scale manifestation of small-scale heterogeneity [e.g., Flügel 1995; Tague and Band 2004]. For example, Newman et al. [2014] spatially integrate across a small number of discrete landscape types in order to reproduce the local runoff "hotspots" described by Seyfried et al. [2009]. More recently, Chaney et al. [2016a] demonstrate that the use of spatially interacting hydrologic response units can reduce computational cost of a fully distributed hydrologic model by three orders of magnitude without appreciable losses in information. Like the large-scale flux parameterizations, there is a strong need to rigorously compare different approaches to explicitly resolve dominant processes.

An interesting twist is the interplay between explicitly representing small-scale processes and avoiding or reducing redundant calculations across large model domains. For example, in the push for hillslope-resolving models across large geographical domains, one approach is to use the concept of representative hillslopes [Troch et al. 2003; Hazenberg et al. 2015; Ajami et al. 2016]. The representative hillslope has a length dimension much smaller than the length scale of the model element, and the hillslope is discretized into columns along an axis perpendicular to the stream to explicitly resolve lateral flow processes. The hydrologic and energy fluxes from the single hillslope, or averaged across local hillslopes of different types, are then considered representative of the model element as a whole. This approach spatially integrates both along a hillslope and among hillslopes. Such multi-scale approaches show considerable promise and will likely be increasingly used to represent how small-scale heterogeneities, interactions among processes and the connectivity of water across the landscape affects large-scale behavior.

A broader challenge is to simulate the myriad controls on catchment evolution, e.g., to predict how energy gradients dictate landscape evolution, how natural selection favors plants that make optimal use of available resources, and how the dynamic interactions between humans and the environment shapes the storage and transmission of water across the landscape [Rodríguez‐Iturbe et al. 1992; Eagleson 2002; Schymanski et al. 2009; Schymanski et al. 2010; Sivapalan et al. 2012; Harman and Troch 2014; Zehe et al. 2014; Clark et al. 2016; Grant and Dietrich 2017]. Addressing this challenge requires shifting focus from traditional approaches at short time scales where "properties define processes" [Gupta et al. 2012] towards approaches at longer time scales that focus on predicting how

"processes define properties" [Rodríguez‐Iturbe et al. 1992; Eagleson 2002; Harman and Troch 2014]. Importantly, it requires treating humans as an endogenous component of the Earth system [Sivapalan et al. 2012; Clark et al. 2015a].

There are of course multiple possible approaches available to simulate dominant hydrologic processes. A useful path forward is to isolate and scrutinize alternative modeling approaches to represent scaling and heterogeneity. Peters-Lidard et al. (this issue) propose the idea that the approximations in our models can be treated as hypotheses that can be tested in an information-based framework. Such advances in model evaluation methods will be critical in order to accelerate advances in process-based hydrologic models.

## 3    Model parameters

### 3.1    Modeling challenges
The second question posed by Freeze and Harlan [1969] focuses on the availability of data to define system properties (model parameter values).

A key part of this modeling challenge revolves around the availability and quality of spatial information on model parameters. For some model parameters, spatial information is not readily available. Examples of missing parameters include those that define the temporal decay of snow albedo and the recession characteristics of shallow aquifers. In such situations process-based hydrologic and land models often treat these uncertain parameters as physical constants, adopting hard-coded parameters that are selected based on order-of-magnitude considerations or on limited experimental data [Mendoza et al. 2015; Cuntz et al. 2016]. For other parameters the available spatial information is limited to broad landscape characteristics; e.g., the parameters controlling carbon assimilation and stomatal conductance are typically tied to vegetation type [Bonan et al. 2011; Niu et al. 2011], or the available soil maps impose the same hydraulic properties over large areas [Miller and White 1998]. Such ill-defined information on vegetation and soils greatly underestimates the tremendous spatial heterogeneity that occurs in nature. Finally, when spatial information does exist it may have limited spatial representativeness and relevance – for example, the information on hydraulic conductivity from soil pits may only have weak relations with the transmission of water throughout catchments [Beven 1989].

Such limitations notwithstanding, the challenge, really, is to make the most of the information we do have, and generate new information where we can (e.g., new observations), in order to improve estimates of the spatial variations in the storage and transmission properties of the landscape, including the scale dependence of these properties and their transferability across spatio-temporal scales [Klemeš 1986; Samaniego et al. 2010; Melsen et al. 2016]. The next section summarizes how the hydrologic modeling community is rising to this challenge.

### 3.2    Modeling solutions
The solutions to improve information on model parameters are general and can be applied across multiple models of different type and complexity. We see three specific paths forward.  First, there are numerous opportunities to improve information on geophysical properties, including estimates of vegetation structure [Simard et al. 2011], soil depth [Pelletier et al. 2016], soil properties [Chaney et al. 2016b], bedrock depth and permeability [Fan et al. 2015] and the physical characteristics of rivers [Gleason and Smith 2014].

Second, it is possible to improve the way that geophysical information is used to estimate model parameters. For example, the Multi-scale Parameter Regionalization (MPR) approach

of Samaniego et al. [2010] focuses attention squarely on the transfer functions that relate geophysical attributes to model parameters – Samaniego et al. apply transfer functions at the finest spatial scale of the geophysical data (e.g., the soil polygons) and then apply parameter-dependent operators to upscale the fine-scale model parameters to the resolution of the model. The parameter estimation in MPR is hence centered on the coefficients in the transfer functions used to relate geophysical attributes to model parameters, maximizing the information extracted from the geophysical data. Much research has focused on pedotransfer functions to relate soil properties to soil parameters [e.g., Schaap et al. 2001; Soet and Stricker, 2003], and there has been limited work to relate geophysical attributes to other model parameters such as those controlling the impact of soil moisture on saturated areas [Balsamo et al. 2011].

Third, there is considerable scope to improve the way that multivariate data is used to constrain model parameter values. A key path forward is to identify different signatures from the data that can be used to improve parameter values in different parts of the model [Gupta et al. 2008; Yilmaz et al. 2008; Pokhrel et al. 2012; Vrugt and Sadegh 2013; Rakovec et al. 2015]. For example, Troy et al. [2008] use regionalized estimates of the runoff ratio to constrain the VIC model at the grid scale, and there is much more that can be done using such methods [e.g., see the approach of Yadav et al. 2007]. In the distributed model context, signatures related to energy and moisture fluxes may now be constrained by remote sensing imagery, e.g. of skin temperature or ET, though this strategy is far from common today. Similarly, remotely sensed estimates of surface water levels [Revilla-Romero et al. 2016] and total basin storage [Tangdamrongsub et al. 2015] could be used as well as reported statistics on water withdrawal [Wada et al. 2014]. Together, focused effort on improving geophysical information, improving the links between geophysical information and model parameters, and better constraining model parameters, will go a long way to improve parameter values across multiple models.

A very different solution is stochastic modeling (e.g., see Kim et al., 1997). Stochastic modeling accepts that many parameters are impossible to measure or estimate, and instead generates synthetic model parameter fields using probability distributions with assumed length scales. For example, Maxwell and Kollet [2008] use spatially correlated random fields of saturated hydraulic conductivity to define the fine-scale spatial structure of their model domain, and evaluate the impact of this fine-scale structure on hillslope runoff. Similar approaches were used by Kollet et al. [2010] in their proof-of-concept study illustrating the spatial integration of fine-scale 3D variably saturated flow simulations. These approaches derive from the indirect upscaling methods (numerical integration across space) developed in stochastic subsurface hydrology. The downside of such stochastic simulation approaches is that multiple realizations are necessary to separate the signals from the imposed random variability, making such approaches computationally challenging for fine-scale simulations over large geographical domains [Fatichi et al. 2016].

A major challenge is to parameterize the deeper subsurface at regional to continental scales in order to support large-domain groundwater modeling [Bierkens 2015; Clark et al. 2015a]. Advances in estimating parameters of the subsurface may profit from new technologies. For example, it will be possible to use monitoring and exploration technologies (e.g., geophysics) to generate ensembles of hydraulic conductivity fields. Once these fields are estimated at high resolution, MPR could be used to estimate effective hydraulic conductivity values to characterize the required subsurface parameters. Also, stochastic methods need to be extended to capture the large structural variability in the

formations and layers that dominate continental domains [Baroni et al. 2017; Schalge et al. 2017].

Recent attempts to parameterize the sub-surface are a good first step. These include maps of global permeability and porosity for the upper 50 m of the world's aquifers [Gleeson et al. 2014], soil characteristics and regolith thickness [Pelletier et al. 2016; Shangguan et al., 2016; Hengl et al., 2017] and global thickness of the upper aquifers [De Graaf et al. 2015; Fan et al. 2015; Fan 2016]. However, these datasets have been globally extrapolated from locally established empirical relationships between subsurface properties and surface lithology [Hartmann and Moosdorf 2012]. None of these approaches resolve the multi-layer structure of aquifers and aquitards. As a consequence, they provide useful information on the interaction between groundwater and evaporation, but have limited use for resolving true hydrogeological challenges such as assessing global groundwater depletion, groundwater age and land subsidence related to groundwater pumping. Concerted efforts are needed to compile a global hydrogeological multilayer model based on national geological maps and archives and local- and regional scale groundwater modelling studies, providing the rich information on the subsurface that already exists for soils.

## 4    Model execution (computing)

### 4.1    Modeling challenges
In their final question Freeze and Harlan [1969] ask if the computer limitations of storage capacity and speed of computation have been overcome, and if their blueprint for process-based hydrologic modeling is now economically feasible. Interestingly, we have made substantial (and economically feasible) advances in computing, yet we have also pushed beyond what they could envision with model resolution and process complexity. As a result, computing remains, ironically, a present-day challenge, and we still routinely push available computing resources to their limit [Kollet et al. 2010; Wood et al. 2011]. We still struggle with tradeoffs among process complexity, spatial complexity, domain size, ensemble size, the time period of the model simulation. We also still struggle to run our most complex models for a large number of model configurations, for example, experimenting with different model parameter sets, different process parameterizations, and different spatial architectures. To answer Freeze and Harlan's question: The computing limitations have not been overcome.

The challenge is as follows: As we push our models to their computational limit, the expense of these complex configurations may permit only a single deterministic simulation for a short time period [e.g., Maxwell et al. 2015; Fatichi et al. 2016]. Such preferences for complexity and large-domain simulations arguably sacrifice opportunities for model analysis, model improvement, and uncertainty characterization. Complex models may struggle with physical realism because computational limitations constrain capabilities to identify and resolve model weaknesses. Paradoxically, more complex models may achieve less physical realism than computationally frugal alternatives. This is a critical concept – though a counter-intuitive one – that ideally should guide the development of new model applications.

### 4.2    Modeling solutions
There are several solutions to these computational challenges, all of which are now being advanced by leading process-based hydrologic modeling groups. The first solution, and the most obvious, is to exploit advances in massively parallel (e.g., exa-scale) computation [Kollet et al. 2010; Wood et al. 2011; Paniconi and Putti 2015; Fatichi et al. 2016]. This

solution is often implemented by running a complex model for the finest grid resolution possible over the domain of interest [e.g., Maxwell et al. 2015; Maxwell and Condon 2016]. A key reason for conducting such spatially resolved simulations is to understand explicit spatial controls on hydrologic processes. For example, Maxwell and Condon [2016] use high resolution continental-domain ParFlow simulations to understand the controls of groundwater flow on the partitioning of evapotranspiration into bare soil evaporation and transpiration.

A second (related) solution to the computing challenge is to improve numerical solvers. In simpler models the need for robust numerical methods is often undervalued, and numerical errors in simple models contaminate model analysis and complicate model calibration [Kavetski et al. 2006b; Kavetski and Clark 2010, 2011]. For example, the "pits" in model parameter surfaces have been shown to be an artifact of numerical solution methods, requiring development of elaborate and time-consuming parameter estimation strategies that are not necessary in models with robust numerical solutions [Kavetski et al. 2006a; Clark and Kavetski 2010; Kavetski and Clark 2010]. In more complex models, advances in solution methods are an active area of research, with several recent advances in numerical solvers and parallelization strategies [Qu and Duffy 2007; Kumar et al. 2009; Kollet et al. 2010; Maxwell 2013]. Across all models there is a need to improve numerical solution methods, e.g., evaluate accuracy-efficiency tradeoffs, to support efficient model analysis and calibration strategies.

A third solution to the computing challenge is to identify model configurations that avoid redundant calculations while still capturing dominant processes. This can be accomplished using the concept of hydrologic similarity, i.e., recognizing that there is no need to repeat calculations for areas of the landscape with very similar forcing and geophysical properties [e.g., Flügel 1995; Tague and Band 2004]. As noted earlier, recent applications of hydrologic similarity methods have shown that it is possible to reduce run times by two to three orders of magnitude, without any loss in information content [Newman et al. 2014; Chaney et al. 2016a]. Also, hydrologic similarity concepts can be effectively applied using multi-scale methods to resolve the dominant spatial gradients that drive flow; for example, using representative hillslopes to explicitly resolve lateral flow processes [Troch et al. 2003; Berne et al. 2005; Hazenberg et al. 2015; Ajami et al. 2016]. In exploring these solutions, we recognize that there is not necessarily a tradeoff between physical realism and computational efficiency – the linkage between spatial complexity and process complexity may be rather weak, as models run using a large number of spatial elements may still miss dominant processes [e.g., Hartmann et al. 2017]

A fourth solution to the computing challenge, especially the concern that the computational cost of complex models sacrifices opportunities for analysis, is to focus on improving model analysis methods. Analysis of complex models is possible by developing surrogate models, i.e., models that emulate the behavior of complex models and run very quickly [Razavi et al. 2012]. Analysis of complex models is also possible through computationally frugal model analysis methods that require a fewer number of model simulations [Rakovec et al. 2014; Hill et al. 2015]. A way to support these types of methods is to use quasi-scale invariant parameterizations (e.g., MPR) to estimate transfer function parameters at coarser resolutions instead of using a high-resolution model setting. Since parameters obtained with the MPR technique are transferable across scales without significant performance loss, models can be applied at higher spatial resolutions as shown by Kumar et al. [2013]. This alternative would lead to computationally efficient large-scale hydrologic predictions and allows performing parameter estimation over large domains.

In short, solving computing challenges will require judiciously combining emerging computing capabilities, advanced numerical methods, justifiable model simplifications, and extensive use of computationally frugal model analysis methods.

## 5    Summary and next steps

In this paper, we review key advances in process-based hydrologic models. We see that the community has risen to meet major hydrologic modeling challenges in diverse and productive ways. Specifically, the community has made noteworthy advances in improving mathematical descriptions of hydrologic processes, in parameter estimation, and in identifying justifiable model simplifications that make more effective use of available computing resources. Many of these modeling advances are general, and can be applied across multiple models of different type and complexity.

To summarize, there are three general opportunities to improve the physical realism of hydrologic models. First, there is still considerable scope to improve mathematical descriptions of hydrologic processes. A major research challenge is the scaling/closure problem, i.e., to represent how small-scale heterogeneities shape large-scale fluxes, interactions among processes, and the connectivity of water across the landscape. While the hydrological modeling community has made progress in this challenge, through statistical-dynamical models, stochastic upscaling theory, scale-appropriate flux parameterizations, and spatial integration across discrete landscape types, much work is still required both to develop new closure schemes and to systematically compare existing modeling approaches. Second, there is considerable scope to improve information on model parameter values and their associated uncertainties. Advances in parameter estimation will require focused effort to improve the available geophysical information (e.g., through improved observations), improve the links between geophysical information and model parameters, and advance methods to use multivariate data to constrain model parameter values. Third, there is a strong need to more effectively use the available computing resources. We argue here that in addition to exploiting advances in massively parallel computation and numerical solution methods, we can also make much more effective use of the available computing through more efficient/agile models (e.g., use of hydrologic similarity concepts). More effective use of available computing resources can increase capabilities for model analysis and uncertainty characterization, and shine the light toward further model improvements.

We see several specific needs underlying these general research themes (see Figure 1 for the general framework):

1. We need to improve the theoretical underpinnings of our hydrologic models [Clark et al. 2016]. Most discussions of inter-model differences focus on a discussion of algorithms rather than a discussion of processes. While there have been some calls in the past to improve the "dialog" between experimentalists and modelers [Seibert and McDonnell 2002], e.g., to focus more on processes, much of the interaction between experimentalists and modelers is focused on individual watersheds [e.g., Tromp-van Meerveld and Weiler 2008; Hopp and McDonnell 2009]. More work is needed to synthesize process explanations from research watersheds to develop more general theories of hydrologic processes [e.g., Tetzlaff et al. 2009], and test these alternative process descriptions in models.

2. We need to expand our prominence in community hydrologic modeling [Wood et al. 2005; Weiler and Beven 2015], both by providing accessible and extensible modeling tools, and also providing key research datasets and model test cases to

scrutinize alternative modeling approaches. Such community activities will result in greater engagement of field scientists in model development and greater collaboration across diverse modeling groups, resulting in substantial improvements in the physical realism and predictive capabilities of hydrologic models. Advancing such community activities requires that we are much more effective and efficient in sharing data and model source code. This goes beyond just by making models and data publicly available, but, critically, integrating models and data in widely-used analysis frameworks and developing model standards to simplify the sharing of source code in models developed by different groups [Clark et al. 2015b; Clark et al. 2016].

3. We need to systematically and comprehensively explore the benefits of competing modeling approaches [Clark et al. 2015a; Clark et al. 2015b; Clark et al. 2016]. A key need is to systematically evaluate information gains/losses using models of varying complexity, exploring the interplay between changes in process complexity and changes in spatial complexity. These assessments will help identify useful model configurations for specific applications. Another need is to scrutinize models using data from research watersheds, both using data on internal states/fluxes and inter-variable relationships, in order to understand the benefits of competing process parameterizations. More generally, and as emphasized by Peters-Lidard et al. [2017], it is important to use applications of information theory to quantify how effectively models use the available information, i.e., to provide an estimate of system predictability, and identify opportunities to improve models.

4. We need to substantially advance the development of new modeling approaches that simulate the temporal dynamics of environmental change. Key challenges include predicting how energy gradients dictate landscape evolution, how natural selection favors plants that make optimal use of the available resources, and how the dynamic interactions between humans and the environment shapes the storage and transmission of water across the landscape [Rodríguez‐Iturbe et al. 1992; Eagleson 2002; Schymanski et al. 2009; Schymanski et al. 2010; Sivapalan et al. 2012; Harman and Troch 2014; Zehe et al. 2014; Clark et al. 2016; Grant and Dietrich 2017].

5. We must advance research on process-oriented approaches to estimate spatial fields of model parameters. The challenge is to estimate spatial variations in the storage and transmission properties of the landscape. Advances are possible through developing new data sources on geophysical attributes [Simard et al. 2011; Gleason and Smith 2014; Fan et al. 2015; Chaney et al. 2016b; Pelletier et al. 2016; De Graaf et al. 2017], new approaches to link geophysical attributes to model parameters [Samaniego et al. 2010; Kumar et al. 2013; Rakovec et al. 2015], and new diagnostics to infer model parameters [Gupta et al. 2008; Yilmaz et al. 2008; Pokhrel et al. 2012]. Such focus will give the parameter estimation problem the scientific attention that it deserves, rather than the far-too-common approach where parameter estimation is relegated to a "tuning exercise" in model applications. This focus on parameter estimation is necessary to improve the physical realism and applicability of process-based models.

6. We need to obtain better data on hydrologic processes. Field campaigns to obtain new data to understand hydrologic processes are less supported and supportable than before [Tetzlaff et al., 2017], thus a key need is to motivate and design new field experiments to advance understanding of the terrestrial component of the

water cycle across scales and locations. Such work is critical to ensure model development is not unduly constrained by the limited experimental field data that we have at present.

7. We need to advance methods for model analysis, especially for complex models. As mentioned above, analysis of complex models is possible by both (a) developing surrogate models, i.e., models that emulate the behavior of complex models and run very quickly [Razavi et al. 2012]; and (b) applying computationally frugal model analysis methods that require a fewer number of model simulations [Rakovec et al. 2014; Hill et al. 2015]. These advances in model analysis are important because complex models are typically calibrated or analyzed using semi-manual or manual strategies, largely due to their immense computational cost (it is only possible to run a handful of simulations). We have very little insight into process/parameter dominance and process/parameter interactions in very complex models, however such information is desperately needed in order to inform meaningful parameter estimation strategies.

8. Finally, and most importantly, we need to improve the construction of hydrologic models. Many of today's models have developed somewhat of a "shantytown" appearance, where a succession of students and post-docs bolted on new components to suit the needs of their particular project, and the overall construction of the model has become rather messy. Clark et al. [2015b] define some key requirements as: (a) impose modularity at the level of the individual fluxes, to enable greater model extensibility and code reuse, as it is straightforward to combine different flux parameterizations to form alternative conservation equations; (b) separate the physical processes from their numerical solution, to enable experimenting with alternative numerical solution methods, e.g., evaluating accuracy-efficiency tradeoffs; and (c) use hierarchal data structures, to enable representing spatial variability and connectivity across a range of spatial scales. Such improvements in model construction are a critical underpinning activity that is critical to accelerate advances in hydrologic science.

In addressing these research tasks it is important to take a unified perspective – it is important to deliberately depart from previous debates on the "correct" approach to hydrologic modeling, and focus instead on more effective use of the diversity of modeling tools to advance our collective quest for physically realistic hydrologic models.

## Acknowledgements

We thank John Ding, Murugesu Sivapalan, Thorsten Wagener, Eric Wood, and an anonymous referee for their constructive comments on an earlier version of this manuscript.

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

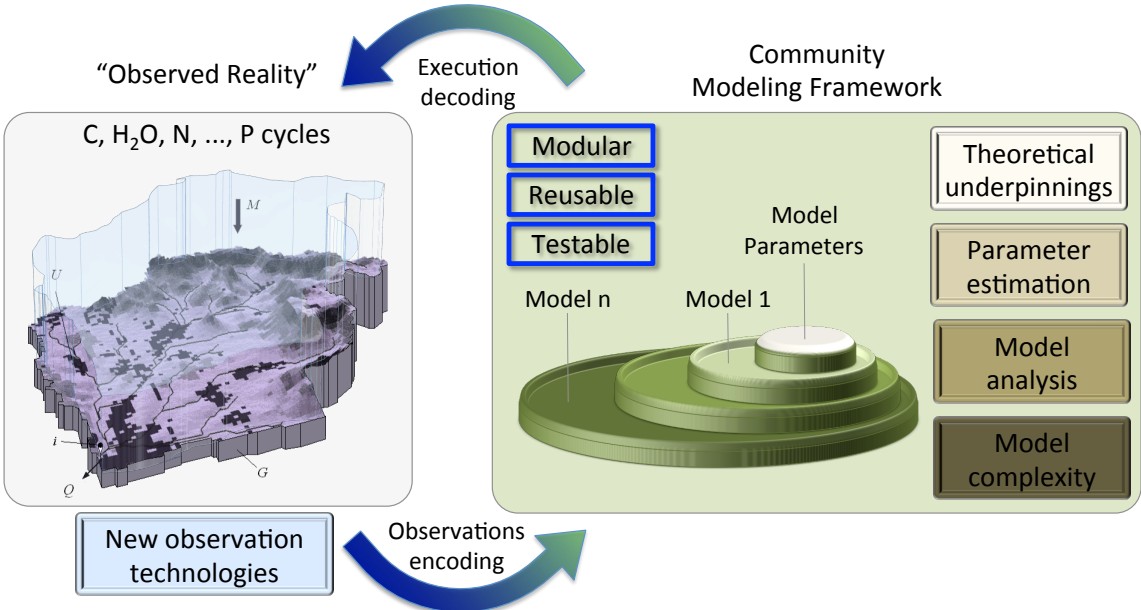

Figure 1. Schematic summarizing some key research priorities to advance the physical realism of process-based hydrologic models.