# Peer review of "The evolution of process-based hydrologic models: Historical challenges and the collective quest for physical realism"

_Hydrology and Earth System Sciences, 2016_

## Short Comment (SC1) · 19 Jan 2017

An engineer's retrospective

In the eyes of the authors, modern process–based hydrology may have started with the blueprint of Freeze and Harlan (1969). But applied hydrology, of which I've had some familiarity, also had similar process-based beginnings.

One standard text I consulted from time to time was one by Chow (1959). Discussed in this, though strictly not a hydrology text, were some hydrologic techniques still in use

these days, such as Horton's and Izzard's overland flow, and the Muskingum channel routing (Chow, 1959, pp.543–549, 604–608).

An earliest synthesis or integration of these surface–water processes had led, in 1967, to a nonlinear Muskingum–storage–type response function as follows (Ding, 2011, Eq. (4), and references cited therein):

$$Q = C^N S^N - C_1(dS/dt), \quad N > 0, \ C > 0, \ 1 \geq C_1 \geq 0.$$

in which $Q$ is the discharge (L/T), $S$ is the stored water (L), $t$ is time (T), $N$ is an exponent (–), $C$ is a scale parameter ($(L/T)^{1/N}$/L), and $C_1$ is a weight (–).

The embedded $(dS/dt)$–term reflects the mass balance or closure for a control volume. The storage–discharge relation as parameterized above may be considered an energy closure for a subsurface flow system as well (e.g. Ibrahim and Brutsaert, 1965, and discussion by Ding (1966) therein; Hammond and Han, 2006, Eq. (3), and especially it's linearized exact solution, Eq. (7)).

Could this storage–driven response function be the elusive "Holy Grail" of scientific hydrology that Beven (2006) and others have been searching for? (See Page 2, Lines 72–76.)

References

Chow, V. T. : Open-channel hydraulics, McGraw-Hill, New York, 1959.

Ding, J. Y.: A measure of watershed nonlinearity: interpreting a variable instantaneous unit hydrograph model on two vastly different sized watersheds, Hydrol. Earth Syst. Sci., 15, 405-423, doi:10.5194/hess-15-405-2011, 2011.

Hammond, M. and Han, D.: Recession curve estimation for storm event separations. Journal of Hydrology, 330(3), pp.573-585, 2006.

Ibrahim, H. A., and Brutsaert, W.: Inflow hydrographs from large unconfined aquifers, J.Irrig.Drain.Div., Am.Soc.Civ.Eng., 91(IR2), 21-38, 1965.

---

## Referee Comment (RC1) · Anonymous Referee #1 · 22 Feb 2017

**Manuscript ID: hess-2016-693**

The evolution of process-based hydrologic models: Historical challenges and the collective quest for physical realism

Clark et al.

Summary ——- The manuscript is one in a series of discussion papers by the lead author over the last few years and was prompted by the Symposium in Honor of Eric Wood: Observations and Modeling across Scales held June 2-3, 2016 in Princeton, NJ. The authors use three questions posed by Freeze and Harlan [1969] to examine progress in process-based hydrologic modeling over the last fifty years and to define

outstanding research challenges. The manuscript is generally well-written, but is more narrow in focus than its title and introduction suggest. The specific answers to the three questions are not based on general surveys of the field, but are provided through the lens of the symposium topics. This is probably a good thing, since it constrains the length of the manuscript, but it requires some rewriting or additional wording to clearly define how the questions from Freeze and Harlan are evaluated in the rest of the manuscript.

Comments — 1. For example, the first question discusses whether physicallybased mathematical descriptions of hydrologic processes are available and whether the relationships between the component phenomena are well enough understood. This question is addressed in section 2 (model structure) by focusing on scaling relationships and the representation of local processes in regional models. While an important aspect of model representation and one that continues to challenge the community, it is not the sole challenge to the question posed by Freeze and Harlan (but an obvious angle given the topic of the symposium). The manuscript would be improved by a better connection between the question from Freeze and Harlan and the discussion of scaling. Explain why scaling is the main topic that is being discussed and how it relates or ranks compared to other aspects of question one. Following that, the discussion of scaling in section 2 can remain largely unchanged.

2. Same for model parameters (section 3). The problem is not only that spatial information is not always of sufficient resolution and quality or that the information does not exist, but also that some parameters are not directly observable at the scale of the application (it's not even always clear that the equations we use at certain spatial scales are necessarily the right ones). The challenge (I.229 and following) is not only to make the best use of the information that we have, but there may also be an opportunity to change our physical descriptions to make better use of the available information at a particular scale. Section 3 focuses mostly on parameter upscaling and does not discuss the use of new data sources, the use of inverse methods, etc. I am not advocating to discuss all topics because the manuscript will lose focus, but it would be good to motivate better why the authors focus on this particular aspect of question 2 from Freeze and Harlan.

3. The section on model execution (section 4) requires some discussion of the purpose or end goal of our simulations. One could argue that we run at higher resolutions and finer time steps simply because we can and because we lack the scaling relationships that allow us to be more "economical" with computing resources. The statement (I.320-321) that "[...] more complex models may not have as much physical realism as computationally frugal alternatives" raises the question 'why are we doing them?'. And the final paragraph of section 4.2 leaves me again wondering what the end goal is of the model simulations.

4. I.389-I.399: It would be nice to call out some of the specific advances that have been made in response to the questions from Freeze and Harlan.

Minor comments — \* I.101-102: "both sets of solutions can occur in the same model". Do you mean for the same process? This is not entirely clear and an example would be helpful.

\* I.103-104: "are readily shared among different modeling groups". I don't quite understand what "shared" means in this context (even given the example).

\* I.129: "summarizes recent in developing". Word missing after "recent"

\* I.141: "Another class of methods is" Suggested change "Another class of methods consists of"

\* I.143: "described earlier" Where, I cannot find the earlier reference.

 $^{\ast}$  I.157: "However, if I and D are comparable in scale, this becomes problematic." Explain why.

\* I.284: "for upper 50 m". Suggested change "for the upper 50 m"

СЗ

\* I.289: "on the other hand". Missing a "on the one hand"

\* I.291: "guide the interaction" should read "guide to the interaction"

---

## Referee Comment (RC2) · M. Sivapalan (Referee) · 12 Mar 2017

This paper has many similarities to the accompanying synthesis paper led by Christa Peters-Lidard. In a certain way, the two papers can be considered intellectual twins, being underpinned by the same Newtonian worldview, which has dominated hydrology for much of the last 40 years. Much of the discussion in the paper centers around the classic paper by Freeze and Harlan, which is the very exemplar of the Newtonian worldview. There is much that I like in the paper – I do agree that Freeze and Harlan provides a good (common) framework to address some of the current modeling challenges. Also, through the use of this common framework the paper remained internally consistent, and is a good vehicle to organize the many contributions of Eric Wood to

hydrologic modeling across scales.

My comments on this paper, and my challenge to the authors, is also going to be driven by the same intellectual argument that adopted in my commentary on the paper by Peters-Lidard et al. (2017). As in the previous case, I am going to pitch my comments at a philosophical level. I hope other, more detailed aspects of the paper get critiqued by other reviewers.

The words "physical realism" appears in the title. I want to discuss the meaning of "physical realism" with examples from personal experience. My true "hydrological" education (as opposed to education based on traditional reductionist theories, e.g., Richards equation) began when I developed (from scratch) my own continuous water balance model, called LASCAM, for the eucalyptus dominated, Mediterranean catchments of Western Australia, published as a 3-part paper in Hydrological Processes (Sivapalan et al., 1996a,b,c). This was back in the early 1990s. By today's standards the model was not so remarkable – indeed, it has many similarities to the HBV model of Sten Bergstroem, even though I was not aware of HBV at the time. After a lot of (political) resistance during my time in Australia, the model is now (only after I left WA) widely used by government agencies and consultants in the State.

What was remarkable – at that time my ecohydrologic background knowledge was minimal – was that the ET component of the model was very "elaborate": it accounted for at least 6 calibrated parameters to account for root water uptake from three separate compartments of the subsurface. I convinced myself that this was appropriate (i.e., physically realistic) and that the model performed well for the right reasons (e.g., through good comparisons to groundwater levels). However, my understanding of physical realism under the circumstances changed dramatically a few years later when one of my PhD students, Richard Silberstein, along with post-doc Neil Viney and several colleagues from CSIRO Australia did a well planned and executed field experiment in one of the jarrah forest catchments. By usual standards, it cost less than $30,000 in real cash, which was also remarkable. The key results are included in 2 papers published

in Agricultural and Forest Meteorology (Silberstein et al., 2001, 2003).

To cut the long story short (details can be found in the published papers), the study was undertaken during two 2-week periods. One was at the end of a very wet winter (potential ET of the order of 4 mm/day), and the other at the end of very dry summer (potential ET of 11 mm/day), i.e., under very wet and dry atmospheric and soil moisture conditions. Yet, remarkably, measured actual total ET (we measured all the components of ET also) was about 2.5 mm/day (almost constant), every day, for each of the two 14-day periods. Remarkable, also because the results made a fool of me for having 6 parameters in LASCAM to capture ET (what would that mean for equifinality and predictive uncertainty?). Now I know the reason – the eucalyptus trees in Western Australia have the ability to send roots down to 30 meters or more to tap into groundwater, and so do not feel the stress that most conceptual models (which I adopted in LASCAM) think they do. This we confirmed through drilling down to the water table and through soil core analyses: there were roots at 30 meters, and when the water table dropped a few meters between winter and summer, the roots followed the water).

My point is that physical realism goes beyond the "physics" that we know and capture in our models, both conceptual and "physically-based". Physical realism must therefore include the fact that native trees, unlike human-engineered crops, adapt themselves to the environment, like the eucalyptus trees in Western Australia adapt through their deep-rooting strategies to cope with the Mediterranean climate (vast fluctuations of soil moisture storage and groundwater levels). Subsequent work by a later PhD student, Stan Schymanski, in northern Australia, near Darwin, which experiences a tropical climate (contrasting wet and dry periods), confirmed further the adaptation ability of native ecosystems, and that the traditional ET modeling techniques derived from crops do not work there either. However, the adaptation strategy here is different. Here trees (that tap into groundwater) transpire at a remarkably constant rate of 1 mm/day throughout the year. However, during the wet season an understory vegetation develops (i.e., grasses) that can transpire at as much as 3 mm/day while the soil is wet,

and then die/senesce (ET = 0) during the dry season. Ecohydrologists among the authors of this paper can confirm that this story is indeed repeated everywhere around the world – the only difference is that you get different vegetation types and different adaptation strategies. This was brought home to me by a recent paper published by Berghuijs et al. (2014) who applied the same conceptual model applied to over 200 MOPEX catchments across the United States, and interpreted the resulting seasonal water balances in terms of the controls of climate, vegetation and soils.

Now, one must add that natural vegetation not only adapts to the prevailing climate and geology, as demonstrated by Berghuijs et al. (2014), but they also adapt/modify the environment around them. I make this argument briefly to connect to my comment on the previous paper by Peters-Lidard et al. (2017) in that notions of hydrologic similarity as well as model parameterizations must account for the co-evolution of climate, soils and vegetation. Berghuijs et al. tried to accommodate this similarity in a Darwinian sense. The adaptations of vegetation is part of this, and the modification of the environment (e.g., soils) by the vegetation is an extension of this co-evolution. Hubert Savenije and his group have been pushing this line of argument for some time, as part of their Flexi modeling framework. The paper(s) by Gao et al. bring out, again through conceptual modeling exercises (in the same spirit as Berghuijs et al.) that the root zone depth (or a bucket capacity in their models) can be seen as an outcome of the co-evolution with the climate (Gao et al., 2014; Savenije and Hrachowitz, 2017). This is indeed physical realism, more broadly defined than the traditional recourse to Newtonian models, where associated parameters are prescribed externally (and also differently in different places). The benefit of treating these parameters are co-evolutionary is that they are better suited to predictions under change.

I am sorry about this, but this is a long-winded prelude to my main comment on the paper. From the early days of my PhD I have been a great fan of Alan Freeze and his pioneering papers. I learned a lot from his papers about how to do science, how to use models to generate new insights, including creative numerical experiments with

models. The paper by Freeze and Harlan has indeed provided the intellectual framework for much of the hydrological modeling we have undertaken in the last 40 years. It connected the strongly fluid mechanics based (mechanistic) paradigm of Eagleson (1970) to the geoscience thinking that followed, and blossomed with the advent of digital terrain models, fast computers and their visualization capability.

However, in view of the arguments that I have made above, we have to come to the realization that Freeze and Harlan has outlived its usefulness. We need a new generation of models combine aspects of the Newtonian worldview embedded in Freeze and Harlan with a Darwinian worldview that accommodates the fact (the true reality) that, through the actions of vegetation (and now humans), whole catchments are co-evolved, almost "living", things that adapt to the climate and geology, and adapt the environment around them.

To be fair, at the time Freeze and Harlan published their paper, the focus was on modeling of runoff generation processes (and there was much less attention paid to water balance per se). In the subsequent 40 years, starting with Eagleson (1978), we have come to realize that evapotranspiration is an important component of water balance, even to model runoff generation processes correctly, e.g., to accommodate the effects of antecedent conditions. This paper, by focusing on Freeze and Harlan, has totally ignored the most important hydrologic process globally, which is ET. So, in effect, the paper is more about the past than about the future.

By interpreting progress in process-based models of hydrology through the prism of Freeze and Harlan, this paper is completely missing many lessons learned from 40 years of modeling effort, and therefore providing a partial, and biased perspective to new entrants to the hydrologic modeling in the future. I don't know exactly what the right modeling approach should be (there is a lot of debate on this) – I am convinced that to be useful it has to combine elements of the Newtonian and Darwinian worldviews. In more modeling language it must mean a combination of the bottom-up (upward) and top-down (downward) approaches – benefiting from the strengths each of them

bring while overcoming their weaknesses (Sivapalan, 2005). Given the co-evolutionary nature of catchment hydrology, we must abandon the false sense of superiority we give to so-called "physically based" models. Nature does not care what models we cook up to mimic it, nor what names we give them. Nature is nature regardless and physical realism must reflect what actually happens in nature, and not what transpires in the human mind about them, or as Klemês (1986) put it, ". . . the logic of hydrological processes cannot be deduced from algebra".

Given the fact that the authors have decided to use Freeze and Harlan to frame this synthesis paper, and the paper is internally consistent, I don't know how it can be turned around now to address my comments. I will leave it up to the authors: perhaps there can be a discussion of the limitations of Freeze and Harlan in respect of physical realism, and in respect of future modeling of hydrological processes, especially under change.

References cited

Eagleson, P. S. (1970). Dynamic Hydrology, McGraw-Hill, New York.

Eagleson, P. S. (1978). Climate, soil, and vegetation, 1. Introduction to water balance dynamics. Water Resources Research, 14, 705–712.

Klemês, V. (1986). Dilettantism in hydrology: transition or destiny? Water Resources Research, 22, 177S–188S.

Sivapalan, M., J. K. Ruprecht and N. R. Viney (1996). Catchment-scale water balance modeling to predict the effects of land use changes in forested catchments. 1. Small catchment water balance model. Hydrological Processes, Vol. 10, No. 3, pp. 393-411.

Silberstein, R. P., A. Held, T. J. Hatton, N. R. Viney, and M. Sivapalan (2001). Energy balance of a natural jarrah (Eucalyptus marginata) forest in Western Australia. Measurements in spring and summer. Agricultural and Forest Meteorology, Vol. 109, pp. 79-104.

Silberstein, R. P., M. Sivapalan, N. R. Viney, A. Held and T. J. Hatton (2003). Modelling the energy balance of a natural jarrah (Eucalyptus marginata) forest. Agricultural and Forest Meteorology, Vol. 115, No. 201–230.

Sivapalan, M. (2005). Pattern, Process and Function: Elements of a New Unified Hydrologic Theory at the Catchment Scale. Contribution to: Encyclopaedia of Hydrologic Sciences, M. G. Anderson (Managing Editor), Chapter 13 (Vol. 1, Part 1), pp. 193-219, John Wiley & Sons.

Schymanski, S. J., M. Sivapalan, M. L. Roderick, L. Hutley and J. Beringer (2009). An optimality-based model of the dynamic feedbacks between natural vegetation and the water balance. Water Resources Research, Vol. 45(1), W01412, doi: 10.1029/2008WR006841.

Berghuijs, W. R., M. Sivapalan, R. A. Woods and H. H. G. Savenije (2014). Patterns of similarity of seasonal water balance: A window into streamflow variability over a range of timescales. Water Resources Research, Vol. 50(7), pp. 5638–5661, doi:10.1002/2014WR015692.

Gao, H., M. Hrachowitz, S. J. Schymanski, F. Fenicia, N. Sriwongsitanon, and H. H. G. Savenije (2014), Climate controls how ecosystems size the root zone storage capacity at catchment scale, Geophys. Res. Lett., 41, 7916–7923, doi:10.1002/2014GL061668.

Savenije, H. H. G. and M. Hrachowitz (2017). HESS Opinions "Catchments as meta-organisms – a new blueprint for hydrological modelling". Hydrol. Earth Syst. Sci., 21, 1107–1116, doi:10.5194/hess-21-1107-2017.

---

## Referee Comment (RC3) · T. Wagener (Referee) · 28 Mar 2017

REVIEW COMMENTS

This manuscript is an interesting contribution to the on-going community debate on how to advance hydrologic models. Given the nature of this manuscript, I will below remark on the three main areas of the commentary, rather than commenting on individual parts. Hopefully my comments will help to expand the nice discussion in this manuscript even further.

One overall issue that might be stressed more in this commentary is that the three areas outlined (model structure, model parameters and model execution) are interdependent, and that improving one requires advancements in the others. For example, it is difficult to reduce parameter spaces for complex models if computing demands do not allow us to explore such spaces thoroughly in the first place.

Some thoughts on the three areas are written-up below:

[1] Model Structure

One issue to mention here might be that there is a trade-off between our ambition to have models that are flexible enough to produce a high performance when matched against observations, that are parsimonious so that parameter uncertainty is low, and that show a high degree of realism in the sense that they are consistent with reality – often equated with models of higher resolution (Wagener, 2003). These are often seen as conflicting objectives. It might be worth discussing that model realism can be achieved with simpler models, while more complex models can still miss key processes or get key fluxes wrong. One example is the recent paper by Hartmann et al. (2017), which compared a widely used global model (PCR-GLOBE) with a much simpler model (Var-Karst). The latter included subsurface heterogeneity and produced much more realistic recharge estimates for karst regions. Another example might be the lack of preferential flow representation in many otherwise complex models.

So how can we ensure that our models are not missing key processes, while we focus on improving details elsewhere? Maybe the top-down approach discussed in another current commentary by Clark and Hrachowitz is a strategy to approach this problem?

[2] Model Parameters

The authors mention the use of signatures for constraining parameter spaces. I think this part might be worth expanding a bit. Such strategies are still not used regularly for distributed models though some nice examples of their value exist. One such example is the paper by Troy et al. (2008) where Eric and his students/colleagues use runoff ratio to constrain VIC at the grid scale. The resulting parameter estimates are much improved by this process.

So, what information can we use to constrain our hyper-resolution models? This information can come from a range of places. For example, it might be possible to synthesise previous experimental and modelling studies that have focused on individual places to gain a better expected value of flux magnitudes across larger domains, or we might be able to use observed vertical fluxes of moisture and energy as 'weak' constraints to account for scale differences

between measurement and model scales (e.g. both done by Hartmann et al., 2017, in relation to karst recharge). Or we might be able to regionalize signatures as constraints beyond those relevant for streamflow, but maybe relevant for ET or other fluxes/states (e.g. the Troy et al., 2008, strategy). How can we reduce the acceptable output space of a model to reduce parameter uncertainty?

[3] Model Execution

This section is assuming that models will become more complex and therefore computationally more challenging. Models might become more complex because they cover a larger domain or because they have more detailed spatial resolutions. One area of inquiry that therefore requires advancements so that it can serve more complex models are optimization and sensitivity analysis tools. We currently explore the parameter spaces of medium complexity models in great detail – to understand the location of the best parameter sets or to understand dominant controls. However, we regularly find that the most complex models are calibrated or analysed using semi-manual or manual strategies, which suggests that there is a mismatch between the models most in need of powerful tools, and the tools we have at our disposal. Most of our currently available tools fall down when confronted with very large problems, i.e. large parameter spaces.

Computational demands can be reduced if we better understand which (modelled) processes are dominant (at particular times or in particular parts of the model domain) – therefore allowing us to search reduced parameter spaces rather than the very large parameter spaces of these models. Our sensitivity analysis methods are not yet particularly good to understand highly interacting parameter spaces though, which is what we typically encounter in complex models. There is also still a lack of how we effectively merge process understanding with optimization/sensitivity analysis to derive approaches tailored to our complex hydrologic models.

References

Hartmann, A., Gleeson, T., Wada, Y. and Wagener, T. (2017). Enhanced recharge rates by altered recharge sensitivity to climate variability through subsurface heterogeneity. Proceedings of the National Academy of Sciences, 2842 -2847, doi: 10.1073/pnas.1614941114

Troy, T. J., E. F. Wood, and J. Sheffield (2008). An efficient calibration method for continental-scale land surface modeling. Water Resour. Res., 44, W09411, doi:10.1029/2007WR006513.

Wagener, T. (2003). Evaluation of catchment models. Hydrological Processes, 17, 3375-3378.

---

## Referee Comment (RC4) · E. Wood (Referee) · 28 Mar 2017

HESS Review of "The evolution of process-based hydrologic models: Historical challenges and the collective quest for physical realism" by M Clark et al.

This is a synthesis paper for the special HESS issue honoring Eric F Wood. The paper is structured around three modeling "challenges" posed by Freeze and Harlan: (1) define suitable model equations – i.e. process parameterizations, (2) define adequate model parameters --- i.e. the adequacy of data and the resulting uncertainty; and (3) cope with limitations in computing power – computational constraints. The paper is very successful in presenting historical modeling challenges and summarizing various approaches developed over the years to address the challenges, but less successful in offering a more comprehensive vision of moving forward.

The review of the historical progress (and literature) is very comprehensive, and a student wanting to read about land surface modeling could spend a semester reading the paper and selected references, and really learn what has been done.

I have one major comment related to areas 1: nothing is mentioned about the numerical schemes used to solve current LSM – especially those like Noah, VIC, Topmodel, mHM, etc. I think the papers by Dmitri Kavetski (e.g. WATER RESOURCES RESEARCH, VOL. 39, NO. 9, 1246, doi:10.1029/2003WR002122, 2003; or JH 320(1,SI)173-186 <arch 2006.) offers important insights that need to be included. Martyn probably know of other similar works, since he is the lead author on WRR 46, Art W10510, Oct 8, 2010 with Dmitri.

Section 5 (Summary and next steps) was rather disappointing. The three points basically says the challenges remain, without any insights as to potential pathways forward. While the majority of the paper would really help students understand LSM developments over the last 40 years, the last section would offer no idea of where new research should go. To say that the key challenge is best posed by a quote by Wood ("*What modeling experiments need to be performed to resolve the "scale" question and what is the trade-off among model complexity, the physical basis for land parameterizations and observational data for estimating model parameters?*"), given the eminence of the author list, leaves this reviewer somewhat disappointed.

I would recommend that the authors augment this last section by listing potential pathways. Does SUMMA offer a framework for the modeling experiments Wood asks for? Can one develop a virtual reality (with or without SUMMA?), as called out by Wood (Wood, Eric F, Jan Boll, Patrick Bogaart and Peter Troch 2005. The Need for a Virtual Hydrologic Laboratory for PUB, Ch 16 in *Predictions in Ungauged Basins: International Perspectives on the State of the Art and Pathways Forward*. Eds. S Franks, M Sivapalan, K Takeuchi, and Y Tachikawa, IAHS Pub 301, Wallingford, Oxon. pp189-203), to explore "*trade-off among model complexity, the physical basis for land parameterizations and observational data for estimating model parameters*"? So I challenge the eminent authors of this synthesis paper to offer students and younger colleagues 'hints' on ways forward. It would make the paper much more impactful.

---

## Author Comment (AC1) · 30 May 2017

We thank the reviewer for his/her constructive comments.

The referee brings up an interesting point, in that the closure relations (i.e., flux parameterizations) can be parsimonious functions of storage. This approach has been described in some of our other papers, in particular, Clark et al. (2008), Gupta et al. (2012) and Hrachowitz and Clark (2017).

Such bucket-style models are extensively used in engineering hydrology. A key example is the Sacramento model used for streamflow forecasting in the USA.

We describe the bucket-style modeling approach on page 3, and we have modified the

text to refer to Hrachowitz and Clark [2017]. The revised text is

"In bucket-style rainfall-runoff models – at the simplest end of the complexity continuum – the large-scale transmission of water is often defined as a linear (or near-linear) function of water storage (e.g., see the synthesis in Clark et al. [2008] and the review by Hrachowitz and Clark [2017])."

References:

Clark, M. P., A. G. Slater, D. E. Rupp, R. A. Woods, J. A. Vrugt, H. V. Gupta, T. Wagener, and L. E. Hay, 2008: Framework for Understanding Structural Errors (FUSE): A modular framework to diagnose differences between hydrological models. Water Resources Research, 44, doi: 1029/2007WR006735.

Hrachowitz, M., and M. P. Clark, 2017: HESS Opinions: The perceived dichotomy between physically-based and conceptual modelling strategies in hydrology and how we can benefit from their convergence. Hydrology & Earth System Sciences, under review.

---

## Author Comment (AC2) · 30 May 2017

**Response to comments from Reviewer 2 on *"The evolution of process-based hydrologic models: Historical challenges and the collective quest for physical realism"* by Martyn P. Clark et al.**

[Responses are in red font at the bottom each sub-section].

**1 Summary**

The manuscript is one in a series of discussion papers by the lead author over the last few years and was prompted by the Symposium in Honor of Eric Wood: Observations and Modeling across Scales held June 2-3, 2016 in Princeton, NJ. The authors use three questions posed by Freeze and Harlan [1969] to examine progress in process-based hydrologic modeling over the last fifty years and to define outstanding research challenges. The manuscript is generally well-written, but is more narrow in focus than its title and introduction suggest. The specific answers to the three questions are not based on general surveys of the field, but are provided through the lens of the symposium topics. This is probably a good thing, since it constrains the length of the manuscript, but it requires some rewriting or additional wording to clearly define how the questions from Freeze and Harlan are evaluated in the rest of the manuscript.

Thanks for these comments. We have revised the introduction to clarify the scope of the paper. Specifically, in the Introduction we state:

*We do not mean to provide a comprehensive review; rather, we present possible solutions to outstanding modeling problems, focusing attention on the research sphere of Eric F. Wood.*

**2 Specific comments**

1. For example, the first question discusses whether physically-based mathematical descriptions of hydrologic processes are available and whether the relationships between the component phenomena are well enough understood. This question is addressed in section 2 (model structure) by focusing on scaling relationships and the representation of local processes in regional models. While an important aspect of model representation and one that continues to challenge the community, it is not the sole challenge to the question posed by Freeze and Harlan (but an obvious angle given the topic of the symposium). The manuscript would be improved by a better connection between the question from Freeze and Harlan and the discussion of scaling. Explain why scaling is the main topic that is being discussed and how it relates or ranks compared to other aspects of question one. Following that, the discussion of scaling in section 2 can remain largely unchanged.

   We have modified the discussion at the start of section 2 to clarify that we focus on contributions from Eric F. Wood.

2. Same for model parameters (section 3). The problem is not only that spatial information is not always of sufficient resolution and quality or that the information does not exist, but also that some parameters are not directly observable at the scale of the application (it's not even always clear that the equations we use at certain

spatial scales are necessarily the right ones). The challenge (l.229 and following) is not only to make the best use of the information that we have, but there may also be an opportunity to change our physical descriptions to make better use of the available information at a particular scale. Section 3 focuses mostly on parameter upscaling and does not discuss the use of new data sources, the use of inverse methods, etc. I am not advocating to discuss all topics because the manuscript will lose focus, but it would be good to motivate better why the authors focus on this particular aspect of question 2 from Freeze and Harlan.

While we appreciate the limitations in scope, we do discuss the points raised by this reviewer.

Some parameters are not directly observable at the scale of the application: We discuss the challenges as "*when spatial information does exist it may have limited spatial representativeness and relevance – for example, the information on hydraulic conductivity from soil pits may only have weak relations with the transmission of water throughout catchments [Beven 1989].*"

Section 3 focuses mostly on parameter upscaling and does not discuss the use of new data sources, the use of inverse methods, etc: We already discuss new data sources "*[…] there are numerous opportunities to improve information on geophysical properties, including estimates of vegetation structure [Simard et al. 2011], soil depth [Pelletier et al. 2016], soil properties [Chaney et al. 2016b], bedrock depth and permeability [Fan et al. 2015] and the physical characteristics of rivers [Gleason and Smith 2014].*" We also discuss the use of inverse methods "*[…] there is considerable scope to improve the way that multivariate data is used to constrain model parameter values […]*".

As the reviewer notes, it is impossible to discuss everything, and we believe that the additional motivation provided at the start of the paper (i.e., where we state that we simply present examples, and that we do not intend to be comprehensive) is sufficient for the reader to appreciate the limited scope of our paper.

3. The section on model execution (section 4) requires some discussion of the purpose or end goal of our simulations. One could argue that we run at higher resolutions and finer time steps simply because we can and because we lack the scaling relationships that allow us to be more "economical" with computing resources. The statement (l.320-321) that "[…] more complex models may not have as much physical realism as computationally frugal alternatives" raises the question 'why are we doing them?'. And the final paragraph of section 4.2 leaves me again wondering what the end goal is of the model simulations.

Good point. We included additional discussion in the section on model execution:

*A key reason for conducting such spatially resolved simulations is to understand explicit spatial controls on hydrologic processes – for example, Maxwell and Condon [2016] use high resolution continental-domain ParFlow simulations to understand the controls of groundwater flow on the partitioning of evapotranspiration into bare soil evaporation and transpiration.*

4. l.389-l.399: It would be nice to call out some of the specific advances that have been made in response to the questions from Freeze and Harlan.

We appreciate this comment. Specific modeling advances are defined throughout the paper, and we do not see the need repeat the modeling advances in the conclusions. Rather, we have expanded the conclusions to address the comment from Eric Wood to define a path forward for the community.

**3   Minor comments**

- l.101-102: "both sets of solutions can occur in the same model". Do you mean for the same process? This is not entirely clear and an example would be helpful.

  We have revised the text to state that many models include a mix of small-scale and large-scale flux parameterizations (e.g., VIC uses a large-scale parameterization of infiltration, yet relies on small-scale equations to simulate the storage and transmission of water through the soil matrix).

- l.103-104: "are readily shared among different modeling groups". I don't quite understand what "shared" means in this context (even given the example).

  The revision above helps address this issue. Further, we have revised the text to state "*When viewed in this way, the different solutions to the scaling/closure problem can be shared among different modeling groups that employ very different modeling approaches.*"

- l.129: "summarizes recent in developing". Word missing after "recent"

  Fixed. "*[…] recent advances […]*"

- l.141: "Another class of methods is" Suggested change "Another class of methods consists of"

  Suggestion adopted. Thanks.

- l.143: "described earlier" Where, I cannot find the earlier reference.

  Clarified. The text now reads "*Examples of this class of methods include the empirically derived storage-discharge relationships described earlier, where the large-scale transmission of water is often defined as a linear (or near-linear) function of water storage [Ambroise et al. 1996; Clark et al. 2008; Fenicia et al. 2011; Brauer et al. 2014].*"

- l.157: "However, if l and D are comparable in scale, this becomes problematic." Explain why.

  We have modified the text to state "*However, it becomes more difficult to define scale-aware parameterizations if l and D are comparable in scale.*"

- l.284: "for upper 50 m". Suggested change "for the upper 50 m"

- l.289: "on the other hand". Missing a "on the one hand"

  We now simply state: "*[…] these datasets have been globally extrapolated from locally established empirical relationships between subsurface properties and surface lithology [Hartmann and Moosdorf 2012].*"

- l.291: "guide the interaction" should read "guide to the interaction

  We have revised the text to state "*As a consequence, they provide useful information on the interaction between groundwater and evaporation, but have limited use […]*"

---

## Author Comment (AC3) · 30 May 2017

**Response to comments from Reviewer 3 on *"The evolution of process-based hydrologic models: Historical challenges and the collective quest for physical realism"* by Martyn P. Clark et al.**

[Responses are in red font at the bottom each sub-section].

[Sub-headings are added by the authors]

**1   General comment**

This paper has many similarities to the accompanying synthesis paper led by Christa Peters-Lidard. In a certain way, the two papers can be considered intellectual twins, being underpinned by the same Newtonian worldview, which has dominated hydrology for much of the last 40 years. Much of the discussion in the paper centers around the classic paper by Freeze and Harlan, which is the very exemplar of the Newtonian worldview. There is much that I like in the paper – I do agree that Freeze and Harlan provides a good (common) framework to address some of the current modeling challenges. Also, through the use of this common framework the paper remained internally consistent, and is a good vehicle to organize the many contributions of Eric Wood to hydrologic modeling across scales.

My comments on this paper, and my challenge to the authors, is also going to be driven by the same intellectual argument that adopted in my commentary on the paper by Peters-Lidard et al. (2017). As in the previous case, I am going to pitch my comments at a philosophical level. I hope other, more detailed aspects of the paper get critiqued by other reviewers.

We appreciate this thoughtful review. Siva's review exposes one important aspect of our paper that we have clarified – that we focus on the *questions* posed by Freeze and Harlan, rather the framework (blueprint) proposed by Freeze and Harlan. This is an important distinction, because we wish to celebrate the wide range of methods used to address the major questions that motivate our science, rather than the fairly narrow set of methods proposed by Freeze and Harlan.

We understand that some readers could (incorrectly) interpret our paper as advances in application of the Freeze-Harlan blueprint.

We have modified the abstract and the introduction to clarify that we focus on the Freeze-Harlan questions rather than the Freeze-Harlan blueprint.

**1.1   Physical realism should include catchment evolution**
The words "physical realism" appears in the title. I want to discuss the meaning of "physical realism" with examples from personal experience. My true "hydrological" education (as opposed to education based on traditional reductionist theories, e.g., Richards equation) began when I developed (from scratch) my own continuous water balance model, called LASCAM, for the eucalyptus dominated, Mediterranean catchments of Western Australia, published as a 3-part paper in Hydrological Processes (Sivapalan et al., 1996a,b,c). This was back in the early 1990s. By today's standards the model was not so remarkable – indeed, it has many similarities to the HBV model of Sten Bergstroem, even though I was not aware of

HBV at the time. After a lot of (political) resistance during my time in Australia, the model is now (only after I left WA) widely used by government agencies and consultants in the State.

What was remarkable – at that time my ecohydrologic background knowledge was minimal – was that the ET component of the model was very "elaborate": it accounted for at least 6 calibrated parameters to account for root water uptake from three separate compartments of the subsurface. I convinced myself that this was appropriate (i.e., physically realistic) and that the model performed well for the right reasons (e.g., through good comparisons to groundwater levels). However, my understanding of physical realism under the circumstances changed dramatically a few years later when one of my PhD students, Richard Silberstein, along with post-doc Neil Viney and several colleagues from CSIRO Australia did a well planned and executed field experiment in one of the jarrah forest catchments. By usual standards, it cost less than $30,000 in real cash, which was also remarkable. The key results are included in 2 papers published in Agricultural and Forest Meteorology (Silberstein et al., 2001, 2003).

To cut the long story short (details can be found in the published papers), the study was undertaken during two 2-week periods. One was at the end of a very wet winter (potential ET of the order of 4 mm/day), and the other at the end of very dry summer (potential ET of 11 mm/day), i.e., under very wet and dry atmospheric and soil moisture conditions. Yet, remarkably, measured actual total ET (we measured all the components of ET also) was about 2.5 mm/day (almost constant), every day, for each of the two 14-day periods. Remarkable, also because the results made a fool of me for having 6 parameters in LASCAM to capture ET (what would that mean for equifinality and predictive uncertainty?). Now I know the reason – the eucalyptus trees in Western Australia have the ability to send roots down to 30 meters or more to tap into groundwater, and so do not feel the stress that most conceptual models (which I adopted in LASCAM) think they do. This we confirmed through drilling down to the water table and through soil core analyses: there were roots at 30 meters, and when the water table dropped a few meters between winter and summer, the roots followed the water).

My point is that physical realism goes beyond the "physics" that we know and capture in our models, both conceptual and "physically-based". Physical realism must therefore include the fact that native trees, unlike human-engineered crops, adapt themselves to the environment, like the eucalyptus trees in Western Australia adapt through their deep-rooting strategies to cope with the Mediterranean climate (vast fluctuations of soil moisture storage and groundwater levels). Subsequent work by a later PhD student, Stan Schymanski, in northern Australia, near Darwin, which experiences a tropical climate (contrasting wet and dry periods), confirmed further the adaptation ability of native ecosystems, and that the traditional ET modeling techniques derived from crops do not work there either. However, the adaptation strategy here is different. Here trees (that tap into groundwater) transpire at a remarkably constant rate of 1 mm/day throughout the year. However, during the wet season an understory vegetation develops (i.e., grasses) that can transpire at as much as 3 mm/day while the soil is wet, and then die/senesce (ET = 0) during the dry season. Ecohydrologists among the authors of this paper can confirm that this story is indeed repeated everywhere around the world – the only difference is that you get different vegetation types and different adaptation strategies. This was brought home to me by a recent paper published by Berghuijs et al. (2014) who applied the same conceptual model applied to over 200 MOPEX catchments across the United States, and interpreted the resulting seasonal water balances in terms of the controls of climate, vegetation and soils.

Now, one must add that natural vegetation not only adapts to the prevailing climate and geology, as demonstrated by Berghuijs et al. (2014), but they also adapt/modify the environment around them. I make this argument briefly to connect to my comment on the previous paper by Peters-Lidard et al. (2017) in that notions of hydrologic similarity as well as model parameterizations must account for the co-evolution of climate, soils and vegetation. Berghuijs et al. tried to accommodate this similarity in a Darwinian sense. The adaptations of vegetation is part of this, and the modification of the environment (e.g., soils) by the vegetation is an extension of this co-evolution. Hubert Savenije and his group have been pushing this line of argument for some time, as part of their Flexi modeling framework. The paper(s) by Gao et al. bring out, again through conceptual modeling exercises (in the same spirit as Berghuijs et al.) that the root zone depth (or a bucket capacity in their models) can be seen as an outcome of the co-evolution with the climate (Gao et al., 2014; Savenije and Hrachowitz, 2017). This is indeed physical realism, more broadly defined than the traditional recourse to Newtonian models, where associated parameters are prescribed externally (and also differently in different places). The benefit of treating these parameters are co-evolutionary is that they are better suited to predictions under change.

This is a very interesting historical perspective.

The *questions* that we examine in this paper do not exclude solutions related to co-evolution. Our focus is how we describe processes, how we define parameters, and how we cope with limited computing power and limited information. Indeed, we do discuss approaches of "co-evolution" in much of our previous work [Clark et al. 2015; Clark et al. 2016], and much of this discussion is also relevant in the current paper.

In response to this comment, which we appreciate for its expansive perspective, we have included discussion of the evolution of the landscape (vegetation, soils, and geology). We state at the end of Section 2:

> *A broader challenge is to simulate the myriad controls on catchment evolution, e.g., to predict how energy gradients dictate landscape evolution, how natural selection favors plants that make optimal use of the available resources, and how the dynamic interactions between humans and the environment shapes the storage and transmission of water across the landscape [Rodríguez‐Iturbe et al. 1992; Eagleson 2002; Schymanski et al. 2009; Schymanski et al. 2010; Sivapalan et al. 2012; Harman and Troch 2014; Zehe et al. 2014; Clark et al. 2016; Grant and Dietrich 2017]. Addressing this challenge requires shifting focus from traditional approaches of "properties define processes" [Gupta et al. 2012] towards predicting how "processes define properties" [Rodríguez‐Iturbe et al. 1992; Eagleson 2002; Harman and Troch 2014]. Importantly, it requires treating humans as an endogenous component of the Earth system [Sivapalan et al. 2012; Clark et al. 2015a].*

I am sorry about this, but this is a long-winded prelude to my main comment on the paper. From the early days of my PhD I have been a great fan of Alan Freeze and his pioneering papers. I learned a lot from his papers about how to do science, how to use models to generate new insights, including creative numerical experiments with models. The paper by Freeze and Harlan has indeed provided the intellectual framework for much of the hydrological modeling we have undertaken in the last 40 years. It connected the strongly

fluid mechanics based (mechanistic) paradigm of Eagleson (1970) to the geoscience thinking that followed, and blossomed with the advent of digital terrain models, fast computers and their visualization capability.

However, in view of the arguments that I have made above, we have to come to the realization that Freeze and Harlan has outlived its usefulness. We need a new generation of models combine aspects of the Newtonian worldview embedded in Freeze and Harlan with a Darwinian worldview that accommodates the fact (the true reality) that, through the actions of vegetation (and now humans), whole catchments are co-evolved, almost "living", things that adapt to the climate and geology, and adapt the environment around them.

We agree with Siva here – the whole point of our paper is to encourage the community to take advantage of diverse modeling approaches. Just as the Darwinian worldview offers some useful insights (e.g., comparative hydrology to develop new theories on catchment function), the Freeze and Harlan framework also offers useful insights (e.g., use of stochastic methods in complex models to understand scaling behavior). The intent of our paper is to recognize that we are all trying to answer the same questions and to learn from each other so that we can improve how we simulate hydrologic processes.

As just noted, the *questions* that we examine in this paper do not exclude solutions related to co-evolution. We hope that the revisions described above help to clarify this point.

**2   Focusing on Freeze and Harlan ignores ET**

To be fair, at the time Freeze and Harlan published their paper, the focus was on modeling of runoff generation processes (and there was much less attention paid to water balance per se). In the subsequent 40 years, starting with Eagleson (1978), we have come to realize that evapotranspiration is an important component of water balance, even to model runoff generation processes correctly, e.g., to accommodate the effects of antecedent conditions. This paper, by focusing on Freeze and Harlan, has totally ignored the most important hydrologic process globally, which is ET. So, in effect, the paper is more about the past than about the future.

Again, we focus on the *questions* posed by Freeze and Harlan, not their modeling approach, and we consider a wide range of processes (well beyond the processes considered by Freeze and Harlan).  We agree that the some of the particulars of the Freeze-Harlan blueprint are dated with respect to the data and computing resources we now enjoy, but the central questions they pose continue to be highly relevant. We think that the revisions to our paper help clarify that we are interested in so much more than 3D variably saturated flow simulations.

**3   The paper provides a biased perspective**

By interpreting progress in process-based models of hydrology through the prism of Freeze and Harlan, this paper is completely missing many lessons learned from 40 years of modeling effort, and therefore providing a partial, and biased perspective to new entrants to the hydrologic modeling in the future.

Again, by focusing on the questions posed by Freeze and Harlan we focus attention on new insights and new concepts that are needed to bring hydrology forward, including catchment evolution and human-water interactions.

I don't know exactly what the right modeling approach should be (there is a lot of debate on this) – I am convinced that to be useful it has to combine elements of the Newtonian and Darwinian worldviews. In more modeling language it must mean a combination of the bottom-up (upward) and top-down (downward) approaches – benefiting from the strengths each of them bring while overcoming their weaknesses (Sivapalan, 2005). Given the co-evolutionary nature of catchment hydrology, we must abandon the false sense of superiority we give to so-called "physically based" models. Nature does not care what models we cook up to mimic it, nor what names we give them. Nature is nature regardless and physical realism must reflect what actually happens in nature, and not what transpires in the human mind about them, or as Klemês (1986) put it, ". . . the logic of hydrological processes cannot be deduced from algebra".

As should be clear by now, we agree with Siva's central point: The intent of our paper is to bring together diverse perspectives. That is, we recognize that we are all trying to answer the same questions, and we wish to consider diverse modeling approaches (e.g., Newtonian and Darwinian approaches, downward and upward philosophies) so that we can learn from each other and improve how we simulate hydrologic processes.

The Freeze-Harlan framework is just one of many modeling approaches that has merit. Many of the equations in the blueprint depend on the conservation of mass, momentum and energy and the 2nd law of thermodynamics, plus a number of well-established constitutive relationships such as Darcy's law and Manning's equation. These modeling approaches are not suddenly obsolete; they just do not paint the entire picture when we consider life and in particular humans.

We hope that our revisions clarify the intent of our paper.

Given the fact that the authors have decided to use Freeze and Harlan to frame this synthesis paper, and the paper is internally consistent, I don't know how it can be turned around now to address my comments. I will leave it up to the authors: perhaps there can be a discussion of the limitations of Freeze and Harlan in respect of physical realism, and in respect of future modeling of hydrological processes, especially under change.

We discuss the limitations of the Freeze and Harlan framework at many points in the paper. We clarify that we focus on the *questions* posed by Freeze and Harlan, and we broaden discussion of the modeling solutions.

---

## Author Comment (AC4) · 31 May 2017

**Response to comments from Reviewer 4 on *"The evolution of process-based hydrologic models: Historical challenges and the collective quest for physical realism"* by Martyn P. Clark et al.**

[Responses are in red font at the bottom each sub-section].

**1    General**

This manuscript is an interesting contribution to the on-going community debate on how to advance hydrologic models. Given the nature of this manuscript, I will below remark on the three main areas of the commentary, rather than commenting on individual parts. Hopefully my comments will help to expand the nice discussion in this manuscript even further.

One overall issue that might be stressed more in this commentary is that the three areas outlined (model structure, model parameters and model execution) are interdependent, and that improving one requires advancements in the others. For example, it is difficult to reduce parameter spaces for complex models if computing demands do not allow us to explore such spaces thoroughly in the first place.

Yes, good point. We now highlight these interdependencies in the Introduction:

> *We discuss these modeling challenges separately, while recognizing that these modeling challenges are strongly interdependent (e.g., a complex model structure may have large computing demands, restricting the extent to which it is possible to explore alternative model parameter sets). We will touch on these interdependencies in the individual sections of the paper.*

We also expand discussion on the interdependencies among modeling challenges in the individual sections of the paper.

**2    Model Structure**

One issue to mention here might be that there is a trade-off between our ambition to have models that are flexible enough to produce a high performance when matched against observations, that are parsimonious so that parameter uncertainty is low, and that show a high degree of realism in the sense that they are consistent with reality – often equated with models of higher resolution (Wagener, 2003). These are often seen as conflicting objectives. It might be worth discussing that model realism can be achieved with simpler models, while more complex models can still miss key processes or get key fluxes wrong. One example is the recent paper by Hartmann et al. (2017), which compared a widely used global model (PCR-GLOBE) with a much simpler model (Var-Karst). The latter included subsurface heterogeneity and produced much more realistic recharge estimates for karst regions. Another example might be the lack of preferential flow representation in many otherwise complex models.

So how can we ensure that our models are not missing key processes, while we focus on improving details elsewhere? Maybe the top-down approach discussed in another current commentary by Clark and Hrachowitz is a strategy to approach this problem?

Good point. We now include discussion of model tradeoffs in the section on computing:

*In exploring these solutions we recognize that there is not necessarily a tradeoff between physical realism and computational efficiency – the linkage between spatial complexity and process complexity may be rather weak, as models run using a large number of spatial elements may miss dominant processes [e.g., Hartmann et al. 2017]*

**2.1  Model Parameters**

The authors mention the use of signatures for constraining parameter spaces. I think this part might be worth expanding a bit. Such strategies are still not used regularly for distributed models though some nice examples of their value exist. One such example is the paper by Troy et al. (2008) where Eric and his students/colleagues use runoff ratio to constrain VIC at the grid scale. The resulting parameter estimates are much improved by this process.

So, what information can we use to constrain our hyper-resolution models? This information can come from a range of places. For example, it might be possible to synthesise previous experimental and modelling studies that have focused on individual places to gain a better expected value of flux magnitudes across larger domains, or we might be able to use observed vertical fluxes of moisture and energy as 'weak' constraints to account for scale differences between measurement and model scales (e.g. both done by Hartmann et al., 2017, in relation to karst recharge). Or we might be able to regionalize signatures as constraints beyond those relevant for streamflow, but maybe relevant for ET or other fluxes/states (e.g. the Troy et al., 2008, strategy). How can we reduce the acceptable output space of a model to reduce parameter uncertainty?

We have added discussion to the section on model parameters:

*[…] there is considerable scope to improve the way that multivariate data is used to constrain model parameter values. A key path forward is to identify different signatures from the data that can be used to improve parameter values in different parts of the model [Gupta et al. 2008; Yilmaz et al. 2008; Pokhrel et al. 2012; Vrugt and Sadegh 2013; Rakovec et al. 2015]. For example, Troy et al. [2008] use regionalized estimates of the runoff ratio to constrain the VIC model at the grid scale, and there is much more that can be done using such methods [e.g., see the approach of Yadav et al. 2007].  In the distributed model context, signatures related to energy and moisture fluxes may now be constrained by remote sensing imagery, e.g. of skin temperature or ET, though this strategy is far from common today.  Similarly, remotely sensed estimates of surface water levels [Revilla-Romero et al., 2016] and total basin storage [Tangdamrongsub et al., 2015] could be used as well as reported statistics on water withdrawal [Wada et al., 2014].*

**2.2  Model Execution**

This section is assuming that models will become more complex and therefore computationally more challenging. Models might become more complex because they cover a larger domain or because they have more detailed spatial resolutions. One area of inquiry that therefore requires advancements so that it can serve more complex models are optimization and sensitivity analysis tools. We currently explore the parameter spaces of

medium complexity models in great detail – to understand the location of the best parameter sets or to understand dominant controls. However, we regularly find that the most complex models are calibrated or analysed using semi-manual or manual strategies, which suggests that there is a mismatch between the models most in need of powerful tools, and the tools we have at our disposal. Most of our currently available tools fall down when confronted with very large problems, i.e. large parameter spaces.

Computational demands can be reduced if we better understand which (modelled) processes are dominant (at particular times or in particular parts of the model domain) – therefore allowing us to search reduced parameter spaces rather than the very large parameter spaces of these models. Our sensitivity analysis methods are not yet particularly good to understand highly interacting parameter spaces though, which is what we typically encounter in complex models. There is also still a lack of how we effectively merge process understanding with optimization/sensitivity analysis to derive approaches tailored to our complex hydrologic models.

We agree with this sentiment. The original paper discussed both surrogate models and computationally frugal model analysis methods, though did not go into great detail. We have revised the paper to highlight the need for model analysis as a key path forward for the community.

> We need to advance methods for model analysis, especially for complex models. As mentioned above, analysis of complex models is possible by both (a) developing surrogate models, i.e., models that emulate the behavior of complex models and run very quickly [Razavi et al. 2012]; (b) applying computationally frugal model analysis methods that require a fewer number of model simulations [Rakovec et al. 2014; Hill et al. 2015]; and (c) developing multi-scale methods that provide insight into finer time-space scale behavior at only the cost of coarser time-space analysis [Samaniego et al. 2010; Rakovec et al. 2015]. These advances in model analysis are important because complex models are typically calibrated or analyzed using semi-manual or manual strategies, largely because of their immense computational cost (it is only possible to run a handful of simulations). We have very little insight process/parameter dominance and process/parameter interactions in very complex models, and such information is desperately needed in order to inform meaningful parameter estimation strategies.

---

## Author Comment (AC5) · 31 May 2017

**Response to comments from Reviewer 5 on "*The evolution of process-based hydrologic models: Historical challenges and the collective quest for physical realism*" by Martyn P. Clark et al.**

[Responses are in red font embedded throughout the review].

This is a synthesis paper for the special HESS issue honoring Eric F Wood. The paper is structured around three modeling "challenges" posed by Freeze and Harlan: (1) define suitable model equations – i.e. process parameterizations, (2) define adequate model parameters – i.e., the adequacy of data and the resulting uncertainty; and (3) cope with limitations in computing power – computational constraints. The paper is very successful in presenting historical modeling challenges and summarizing various approaches developed over the years to address the challenges, but less successful in offering a more comprehensive vision of moving forward.

Thanks for the constructive comments. We now define a clear path forward at the end of the paper (7 extended bullet points; see the detailed response below).

The review of the historical progress (and literature) is very comprehensive, and a student wanting to read about land surface modeling could spend a semester reading the paper and selected references, and really learn what has been done. I have one major comment related to areas 1: nothing is mentioned about the numerical schemes used to solve current LSM – especially those like Noah, VIC, Topmodel, mHM, etc. I think the papers by Dmitri Kavetski (e.g. WATER RESOURCES RESEARCH, VOL. 39, NO. 9, 1246, doi:10.1029/2003WR002122, 2003; or JH 320(1,SI)173 - 186 <arch 2006.) offers important insights that need to be included. Martyn probably know of other similar works, since he is the lead author on WRR 46, Art W10510, Oct 8, 2010 with Dmitri.

Thanks. We've now included discussion of the numerical solutions. In the section on model execution, we now state:

> *A second (related) solution to the computing challenge is to improve numerical solvers. In simpler models the need for robust numerical methods is often undervalued, and numerical errors in simple models contaminate model analysis and complicate model calibration [Kavetski et al. 2006b; Kavetski and Clark 2010, 2011]. For example, the "pits" in model parameter surfaces have been shown to be an artifact of numerical solution methods, requiring development of elaborate and time-consuming parameter estimation strategies that are not necessary in models with robust numerical solutions [Kavetski et al. 2006a; Clark and Kavetski 2010; Kavetski and Clark 2010]. In more complex models, advances in solution methods are an active area of research, with several recent advances in numerical solvers and parallelization strategies [Qu and Duffy 2007; Kumar et al. 2009; Kollet et al. 2010; Maxwell 2013]. Across all models there is a need to improve numerical solution methods, e.g., evaluate accuracy-efficiency tradeoffs, to support efficient model analysis and calibration strategies.*

Section 5 (Summary and next steps) was rather disappointing. The three points basically says the challenges remain, without any insights as to potential pathways forward. While

the majority of the paper would really help students understand LSM developments over the last 40 years, the last section would offer no idea of where new research should go. To say that the key challenge is best posed by a quote by Wood (" What modeling experiments need to be performed to resolve the "scale" question and what is the trade - off among model complexity, the physical basis for land parameterizations and observational data for estimating model parameters? "), given the eminence of the author list, leaves this reviewer somewhat disappointed.

I would recommend that the authors augment this last section by listing potential pathways. Does SUMMA offer a framework for the modeling experiments Wood asks for? Can one develop a virtual reality (with or without SUMMA?), as called out by Wood ( Wood, Eric F, Jan Boll, Patrick Bogaart and Peter Troch 2005. The Need for a Virtual Hydrologic Laboratory for PUB, Ch 16 in Predictions in Ungauged Basins: International Perspectives on the State of the Art and Pathways Forward . Eds. S Franks, M Sivapalan, K Takeuchi, and Y Tachikawa, IAHS Pub 301, Wallingford, Oxon. pp189 - 203), to explore " trade - off among model complexity, the physical basis for land parameterizations and observational data for estimating model parameters"? So I challenge the eminent authors of this synthesis paper to offer students and younger colleagues 'hints' on ways forward. It would make the paper much more impactful.

*Fair comment. We have revised the conclusions to define a clear path forward:*

[revised manuscript text omitted]